# Effect of Dietary Benzoic Acid Supplementation on Growth Performance, Rumen Fermentation, and Rumen Microbiota in Weaned Holstein Dairy Calves

**DOI:** 10.3390/ani14192823

**Published:** 2024-09-30

**Authors:** Haonan Dai, Qi Huang, Shujing Li, Dewei Du, Wenli Yu, Jia Guo, Zengyuan Zhao, Xin Yu, Fengtao Ma, Peng Sun

**Affiliations:** 1State Key Laboratory of Animal Nutrition and Feeding, Institute of Animal Science, Chinese Academy of Agricultural Sciences, Beijing 100193, Chinaworkhq@163.com (Q.H.); dudw120459@163.com (D.D.); guojia_work@163.com (J.G.); 82101212373@caas.cn (X.Y.); fengtaoma@163.com (F.M.); 2Shijiazhuang Tianquan Elite Dairy Ltd., Shijiazhuang 050200, China; embryochina@163.com (S.L.); anboyuwenli@126.com (W.Y.); zhaozengyuanjsdx@163.com (Z.Z.)

**Keywords:** benzoic acid, growth performance, butyrate, iso-butyrate, *Bifidobacterium*, dairy calf

## Abstract

**Simple Summary:**

Weaning stress affects the health of calves, inhibiting growth and disrupting gastrointestinal function. Benzoic acid (BA) is known for promoting growth and intestinal health. This study aims to investigate the effects of BA on the growth performance, rumen fermentation, and rumen microbiota of weaned Holstein dairy calves. Thirty-two Holstein dairy calves (60 days of age) were randomly divided into four groups (*n* = 8) supplemented with 0%, 0.25%, 0.50%, and 0.75% BA to the basal diet (dry matter basis) for 42 days, respectively. The results showed that supplementation with 0.50% BA promoted growth performance by increasing feed intake and average daily gain and reducing feed-to-gain ratio, improved rumen fermentation parameters, and increased the relative abundance of probiotics in the rumen. This suggests that 0.50% BA supplementation might be an effective method for the efficient breeding of weaned calves.

**Abstract:**

Supplementation with benzoic acid (BA) in animal feed can reduce feeds’ acid-binding capacity, inhibit pathogenic bacterial growth, enhance nutrient digestion, and increase intestinal enzyme activities. This study aimed to investigate the effects of different doses of BA on the growth performance, rumen fermentation, and rumen microbiota of weaned Holstein dairy calves. Thirty-two Holstein calves at 60 days of age were randomly assigned into four groups (n = 8): a control group (fed with a basal diet without BA supplementation; CON group) and groups that were supplemented with 0.25% (LBA group), 0.50% (MBA group), and 0.75% (HBA group) BA to the basal diet (dry matter basis), respectively. The experiment lasted for 42 days, starting at 60 days of age and ending at 102 days of age, with weaning occurring at 67 days of age. Supplementation with BA linearly increased the average daily gain of the weaned dairy calves, which was significantly higher in the LBA, MBA, and HBA groups than that in the CON group. The average daily feed intake was quadratically increased with increasing BA supplementation, peaking in the MBA group. Supplementation with BA linearly decreased the feed-to-gain (F/G) ratio, but did not affect rumen fermentation parameters, except for the molar proportion of butyrate and iso-butyrate, which were linearly increased with the dose of BA supplementation. Compared with the CON group, the molar proportions of iso-butyrate in the LBA, MBA, and HBA groups and that of butyrate in the HBA group were significantly higher than those in the CON group. Supplementation with BA had no significant effect on the alpha and beta diversity of the rumen microbiota, but significantly increased the relative abundances of beneficial bacteria, such as *Bifidobacterium*, and reduced those of the harmful bacteria, such as *unclassified_o__Gastranaerophilales* and *Oscillospiraceae_UCG-002*, in the rumen. Functional prediction analysis using the MetaCyc database revealed significant variations in the pathways associated with glycolysis across groups, including the GLYCOLYSIS-TCA-GLYOX-BYPASS, GLYCOL-GLYOXDEG-PWY, and P105-PWY pathways. In conclusion, BA supplementation improved the composition and function of rumen microbiota, elevated the production of butyrate and iso-butyrate, and increased the growth performance of weaned Holstein dairy calves.

## 1. Introduction

Early weaning without negative effects has been considered to decrease the cost of raising dairy heifer calves [1]. However, the reason that early weaning has not been widely used in most dairy farms is due to the subsequently exacerbated weaning stresses, which affect the health of the calves, inhibiting growth and disrupting gastrointestinal function [1,2]. Therefore, within the context of modern intensive dairy farming, mitigating weaning stress in calves emerges as an urgent imperative [3]. In addition to the determination of the optimum age and appropriate weaning method [1,2], many studies have proved that supplementation with feed additives is an effective strategy that helps alleviate weaning stress [3,4].

Organic acids, characterized by their carboxylic acid group (R-COOH), serve as feed additives and are known for enhancing the growth performance and intestinal health of livestock and poultry. They are extensively incorporated into diets for these animals and are being explored as potential alternatives to antibiotic growth promoters [5,6]. In addition, benzoic acid (BA) is an organic acid compound and the simplest aromatic acid structurally [7,8]. As a monocarboxylic acid, BA exhibits broad-spectrum antibacterial properties and notable effectiveness against various bacteria. Its applications span across diverse industries, including industrial production, cosmetics, pharmaceuticals, food, and feed industries, and it is made into organic acidifiers, fragrances, plasticizers, and preservatives [9,10,11,12,13,14].

Previous research has demonstrated the beneficial effects of BA supplementation in various livestock species. In nursery and grower-finisher pigs, dietary supplementation with 0.3% and 0.5% BA significantly enhances their growth performance [15]. Nursery pigs exhibit a linear response to BA up to 0.5%, while the optimal average daily gain (ADG) in grower-finisher pigs is achieved with a calculated supplementation level of 0.36% [15]. Furthermore, combining BA and *Bacillus coagulans* improves the growth performance and nutrient digestibility in piglets while maintaining a balance of intestinal microflora [16]. In turkeys, adding 0.1% BA to their diet enhances caecal function by increasing lactic acid bacteria populations, reducing coliform bacteria populations, and lowering the buffering capacity and pH values in caecal contents [17]. For laying hens, supplementation with 1000 mg/kg BA significantly improves egg quality without affecting production performance [18]. The BA also promotes intestinal health by improving intestinal morphology and enriching microbial composition [18]. In the context of beef production, adding 0.5% BA in the diet for finishing steers does not affect the shelf life of beef post-slaughter, nor does it impact beef color or lipid oxidation [19]. Similarly, adding 0.5% BA to the diet of fattening cattle does not alter carcass characteristics and performance, but significantly increases eating rate and dry matter intake (DMI) [20].

Considering its efficacy of promoting growth and intestinal health in monogastric animals, we hypothesized that supplementation with BA might potentially promote the growth and improve the rumen fermentation as well as the composition and function of the rumen microbiota of dairy calves during and after weaning. Therefore, this study aimed to investigate the effects of varying doses of BA supplementation on the growth performance, rumen fermentation, and rumen microbiota of weaned Holstein dairy calves. The findings of this study may provide insights into optimizing the rearing practices of weaned calves efficiently.

## 2. Materials and Methods

The animal protocol for this research was approved by the Experimental Animal Ethics Committee of the Institute of Animal Science, Chinese Academy of Agricultural Sciences (IAS, CAAS, Beijing, China) (approval number IAS2023-138). All the procedures involved were carried out in accordance with the “Guidelines for the Management and Use of Laboratory Animals” of the IAS, CAAS.

### 2.1. Animals, Diets, and Experimental Design

A total of 32 Holstein heifer calves (60 days of age) with similar body weight (91.8 ± 0.6 kg) were selected for this study and randomly assigned to 4 groups (*n* = 8) using a random number generator. The control group (CON group) received a basal diet, while the low BA group (LBA group), medium BA group (MBA group), and high BA group (HBA group) were supplemented with 0.25%, 0.50%, and 0.75% BA in the basal diet (as dry matter basis), respectively.

This study was conducted at Xingtai Junchang Pastoral Industry Co., Ltd. (Xingtai, Hebei Province, China). Benzoic acid (100% purity) was purchased from Guangzhou Huayu Biotechnology Co., Ltd. (Guangzhou, China). Newborn calves were fed 4 L colostrum within one hour of birth. Thereafter, 2 L of colostrum was fed three times a day for two consecutive days. At 3 days of age, the calves were transferred from the nursery pens to individual calf hutches (3.7 × 1.4 × 1.5 m, enclosed by iron railings and lined with hay). The heifers were fed with 8 L heated raw milk and starter from 4 days of age. Oat grass had been supplemented to the heifers at 21 days of age. The dairy calves started weaning at 60 days of age, involving a 7-day transitional period during which milk feeding gradually decreased by 1 L per day until its cessation at 67 days of age. Solid feed which meets the growth and nutritional needs of weaned calves from 68 days of age was then provided continuously until the conclusion of the study at 102 days of age. The experimental period lasted for a total of 42 days. During the transition period, the dairy calves were fed milk three times a day (05:00, 12:00, and 19:00), with the addition of starter once (08:30). After weaning, growth feed was added three times a day (08:30, 14:30, and 21:30), and calves had ad libitum access to feed and water. The composition and nutrient levels of the basal diet are shown in Table 1, and meet or exceed the nutritional requirements recommended by the National Research Council (NRC, 2021) [21].

### 2.2. Sample Collection and Analysis

#### 2.2.1. Body Weight and Feed Intake

The initial body weight (IBW) and final body weight (FBW) of the dairy calves were recorded at 60 and 102 days of age, respectively. The ADG was obtained by subtracting the IBW from the FBW and dividing the difference by the experimental period. The amount of leftover feed was recorded every day, which was used to calculate the average daily feed intake (ADFI) and feed-to-gain (F/G) ratio.

#### 2.2.2. Determination of Nutrient Levels in Diet

During the experimental period, feed was collected for analysis every two weeks using the quartering method. The measurements of dry matter (DM, AOAC, 2005; method 930.15) [22], crude protein (CP, AOAC, 2000, method 976.05) [23], and ether extract (EE, AOAC, 2003, method 4.5.05) [24] were carried out according to the standard protocol of the Association of Official Analytical Chemists. The measurements of neutral detergent fiber (NDF) content and acid detergent fiber (ADF) contents were carried out according to the method of Van Soest et al. [25]. The contents of Ca and P were measured by atomic absorption spectroscopy (AOAC International,1990; method 985.35) [26] and spectrophotometry (AOAC International, 1990; method 986.24) [26], respectively.

#### 2.2.3. Rumen Fluid Sampling

At 101 days of age, rumen fluid samples were collected from each dairy calf using an oral cavity collector precisely 2 h before the morning feeding. The initial 30 mL of rumen fluid was discarded to ensure the sample’s integrity. Subsequently, 80 mL of rumen fluid was extracted and filtered through four layers of gauze, then quick frozen in liquid nitrogen and stored in liquid nitrogen for the determination of volatile fatty acids (VFAs) and rumen microbiota.

#### 2.2.4. Determination of VFAs

After thoroughly shaking the rumen fluid sample, 2 mL of the sample was taken and centrifuged at 10,000× *g* and 4 °C for 10 min. The supernatant was collected and added 0.15 mL of 25% metaphosphoric acid for fixation. The concentrations of VFAs were determined using a gas chromatograph (7890B-7000D, Agilent Technologies, Santa Clara, CA, USA) [27].

#### 2.2.5. DNA Extraction, PCR Amplification, and 16S rRNA Gene Sequencing

The total microbial DNA was extracted from rumen fluid samples using the YH Soil DNA Extraction Kit (Meiji Biopharmaceutical Technology Co., Ltd., Shanghai, China), following the manufacturer’s instructions. DNA concentration and purity were assessed using a NanoDrop 2000 (Thermo Fisher Scientific, Waltham, MA, USA) and quality was verified via 1% agarose gel electrophoresis.

The V3-V4 region of the bacterial 16S rRNA genes was amplified using the ABI GeneAmp 9700 PCR thermocycler (ABI, Los Angeles, CA, USA) with the forward primer 338F (5′-ACTCCTACGGGAGGCAGCAG-3′) and reverse primer 806R (3′-GGACTACHVGGGTWTCTAAT-5′) [28]. The PCR reaction was performed in triplicate with a reaction mixture consisting 4 μL 5 × FastPfu Buffer, 2 μL 2.5 mM dNTPs, 0.8 μL forward primer (5 μM), 0.8 μL reverse primer (5 μM), 0.4 μL FastPfu Polymerase, 0.2 μL BSA, 10 ng template DNA, and added ddH_2_O to 20 μL [29]. The PCR cycling conditions comprised initial denaturation at 95 °C for 3 min, followed by 30 cycles at 95°C for 30 s, 60 °C for 30 s, 72 °C for 45 s, and a final extension at 72 °C for 10 min.

Following PCR, the amplicons were extracted using 2% agarose gel and further purified with the AxyPrep DNA Gel Extraction Kit (Axygen Biosciences, Union City, CA, USA). The DNA concentrations were quantified using the QuantiFluor-ST (Promega, Madison, WI, USA). Purified amplicons were pooled in equimolar concentrations and subjected to paired-end sequencing (PE250) on the NextSeq2000 platform (Illumina, San Diego, CA, USA) according to the standard protocols. The 16S rRNA sequencing data for all the samples were deposited into the NCBI Sequence Read Archive (SRA) under accession number PRJNA1128666 (https://www.ncbi.nlm.nih.gov/sra/PRJNA1128666, accessed on 26 June 2024).

Raw sequence tags were quality-filtered using QIIME (version 1.9.1, Denver, CO, USA) to obtain high-quality clean tags, and FLASH (version 1.2.11, Baltimore, MD, USA) was used for splicing. Sequences with low quality (scores < 20), reads containing ambiguous bases or unmatched to primer sequences, and reads < 200 bp were filtered and removed, and the respective barcodes were removed. Operational taxonomic units (OTUs) were generated at ≥97% similarity using UPARSE (version 7.0.1090, San Diego, CA, USA) [30]. Then the representative sequences were taxonomically classified against the SILVA Database (Release 138, Bremen, HB, Germany) using the RDP Classifier (version 2.11, East Lansing, MI, USA). The alpha diversity analyses (Ace, Chao, Coverage, Shannon, Simpson, Sobs) were performed using the Mothur (version.1.30.2, Michigan, MI, USA). The principal co-ordinates analysis (PCoA) was performed using a Bray–Curtis distance matrix [31]. The functional predictive analysis was carried out using PICRUSt2 (version 2.2.0, Cambridge, MA, USA).

#### 2.2.6. Statistical Analysis

The growth performance and rumen fermentation parameters among groups were analyzed using one-way ANOVA followed by Tukey’s test in SPSS software (IBM SPSS statistics 26, Chicago, IL, USA). Differences in the alpha diversity indices and relative abundances of rumen microbiota were carried out using the Kruskal–Wallis H test method and stats package in R software (version 3.3.1, Vienna, Austria). Spearman’s correlation analysis was used to analyze the differential pathways prediction based on the MetaCyc database, and relationships between differential rumen microbiota, differential pathways, and VFAs. Results were presented as the mean ± standard error of the mean (SEM), and statistical significance was defined as *p* < 0.05.

## 3. Results

### 3.1. Growth Performance

As shown in Table 2, the ADFI showed a quadratic increase with BA supplementation, peaking in the MBA group (*p* = 0.01). Compared to the CON group, ADFI was significantly higher in the LBA and MBA groups (*p* = 0.01). Similarly, BA supplementation linearly increased the ADG in weaned dairy calves, with significantly higher values observed in the LBA, MBA, and HBA groups compared to the CON group (*p* = 0.03). Correspondingly, the FBW of the dairy calves also increased quadratically with BA supplementation (*p* = 0.02). In addition, the F/G ratio linearly decreased with the supplementation with BA (*p* = 0.03), but there was no significant difference on treatment (*p* = 0.17).

### 3.2. Rumen Fermentation Parameters

Supplementation with BA did not affect rumen fermentation parameters except for the molar proportions of butyrate and iso-butyrate, which showed a linearly increase with BA dose (*p* < 0.01). Specifically, compared to the CON group, the molar proportion of iso-butyrate was significantly higher in the LBA, MBA, and HBA groups (*p* < 0.01), and that of butyrate was higher in the HBA group (*p* = 0.04). The acetate-to-propionate (A/P) ratio linearly increased with increasing BA supplementation levels (*p* = 0.02), but there was no significant difference on treatment (*p* = 0.07), as shown in Table 3.

### 3.3. Composition of Rumen Microbiota

A total of 2,030,939 high-quality sequences were acquired, with the average of 63,483 ± 6186 sequences per sample which belonged to 439 ± 94 OTU. Figure 1A demonstrated that all samples achieved coverage indices above 99.0%, indicating comprehensive species diversity and community structure. The richness index (Chao, Ace, Sobs, Shannon, Simpson, Sobs) did not significantly differ among groups (Figure 1, *p* > 0.05).

As shown in Figure 2, there was no distinct separation between or within groups (*p* > 0.05).

Taxonomic analysis identified 18 phyla in the rumen fluid, predominantly *Bacteroidetes*, *Firmicutes*, and *Proteobacteria*, which collectively accounted for 96.8% (CON), 97.1% (LBA), 98.3% (MBA), and 97.9% (HBA) of the total phyla (Figure 3A). *Prevotella*, *Prevotella*_*7* and *Erysipelotrichaceae UCG*-*002* were the dominant genera in all groups (Figure 3B), with varying relative abundance: *Prevotella* (33.3% CON, 44.9% LBA, 29.9% MBA, and 28.6% HBA), *Prevotella*_*7* (5.9% CON, 6.8% LBA, 12.2% MBA, and 19.5% HBA), and *Erysipelotrichaceae UCG*-*002* (11.7% CON, 4.9% LBA, 5.6% MBA, and 13.8% HBA).

### 3.4. Comparison of Rumen Microbiota at the Genus Level

At the genus level, 14 genera showed a significant difference in their relative abundance among the CON, LBA, MBA, and HBA groups (Figure 4, *p* < 0.05), spanning the *Bacteroideta*, *Firmicutes*, *Cyanobacteria*, *Actinobacteriota*, and *Proteobacteria* phyla. The BA supplementation reduced the relative abundances of *Shuttleworthia* (*p* = 0.01), *unclassified_o__Gastranaerophilales* (*p* = 0.02), *Oscillospiraceae_UCG-002* (*p* = 0.04), and *Prevotellaceae_UCG-003* (*p* = 0.04) while increasing that of *Bifidobacterium* (*p* = 0.01).

### 3.5. Functional Differences in Rumen Microbiota

Functional profiles constructed using the MetaCyc database revealed nine significantly different pathways related to the biosynthesis of key metabolites in glycolysis and amino acid metabolism among the groups (Figure 5, *p* < 0.05). The BA supplementation down-regulated eight metabolic pathways: P105-PWY, GLYCOLYSIS-TCA-GLYOX-BYPASS, TCA-GLYOX-BYPASS, GLYCOL-GLYOXDEG-PWY, PWY-6728, PWY0-42, PWY-6562, and PWY1G-0.

### 3.6. Correlation Analysis of Differential Pathways, Differential Rumen Microbiota, and VFAs

Spearman’s correlation analysis was conducted to explore the relationships among differential pathways, differential rumen microbiota, and VFAs. As shown in Figure 6A, *Unclassified_o__Gastranaerophilales* showed positive correlations with GLYCOLYSIS–TCA-GLYOX-BYPASS (R = 0.52, *p* < 0.01), P105-PWY (R = 0.47, *p* < 0.01), PWY0-42 (R = 0.49, *p* < 0.01), TCA-GLYOX-BYPASS (R = 0.52, *p* < 0.01), and GLYCOL-GLYOXDEG-PWY (R = 0.44, *p* < 0.05). *Bifidobacterium* was negatively correlated with PWY-5088 (R = −0.39, *p* < 0.05) and PWY-6562 (R = −0.37, *p* < 0.05).

Figure 6B highlighted that iso-butyrate exhibited negative correlations with PWY-5088 (R = −0.42, *p* < 0.05), PWY-6562 (R = −0.37, *p* < 0.05), GLYCOL-GLYOXDEG-PWY (R = −0.50, *p* < 0.01), P105-PWY (R = −0.50, *p* < 0.01), GLYCOLYSIS–TCA-GLYOX-BYPASS (R = −0.58, *p* < 0.001), PWY0-42 (R = −0.57, *p* < 0.001), and TCA-GLYOX-BYPASS (R = −0.58, *p* ≤ 0.001). Butyrate showed negative correlations with GLYCOL-GLYOXDEG-PWY (R = −0.41, *p* < 0.05), P105-PWY (R = −0.44, *p* < 0.05), GLYCOLYSIS–TCA-GLYOX-BYPASS (R = −0.52, *p* < 0.01), PWY0-42 (R = −0.49, *p* < 0.01), PWY-6728 (R = −0.54, *p* < 0.01), and TCA-GLYOX-BYPASS (R = −0.52, *p* < 0.01).

In Figure 6C, it can be seen that iso-butyrate demonstrated a positive correlation with *Bifidobacterium* (R = 0.42, *p* < 0.05) and a negative correlation with *unclassified_o__Gastranaerophilales* (R = −0.58, *p* ≤ 0.001). Butyrate was negatively correlated with *Unclassified_o__Gastranaerophilales* (R = −0.43, *p* < 0.05), *unclassified_o__Clostridia_vadinBB60_group* (R = −0.43, *p* < 0.05), and *Prevotellaceae_UCG-003* (R = −0.48, *p* < 0.01).

## 4. Discussion

Weaning stress may affect the health and wellbeing of dairy calves, reducing their growth performance and gastrointestinal development [4]. The ADG and F/G ratios are related to feed types, animal varieties, and management levels, while ADFI is influenced by the energy content and NDF content of diet, feed palatability, and feeding method [32,33,34,35,36]. Although numerous studies have indicated that supplementation with BA could improve the growth performance of livestock and poultry, few studies have reported the effect of BA on the growth performance of weaned Holstein dairy calves. Diao et al. [37] found that supplementation with 5000 mg/kg BA increased the ADG and ADFI in young pigs. Another study suggested that 10,000 mg/kg BA supplementation significantly improved the growth performance in weaned pigs and significantly decreased their F/G ratio [38]. Józefiak et al. [39] found that 0.1% BA supplantation resulted in an increase in the BWG of chickens in the first 2 weeks of fattening [39]. Although BA has been extensively studied in monogastric animals, research on its effects in ruminants, particularly weaned calves, remains limited. Only Williams et al. [20] found that adding 0.5% BA to the diet of fattening cattle significantly increased eating rate and DMI. The present study showed that the 0.5% BA supplementation significantly improved the ADG and ADFI, and linearly decreased the F/G ratio of weaned Holstein dairy calves, results which were consistent with the conclusions of previous research [20,37,38,39].

Dietary carbohydrates and other plant fibers, after being anaerobically fermented by a variety of bacteria and enzymes in the rumen, are broken down and then transform into VFAs, methane, CO_2_, and hydrogen. It is known that ruminal VFAs can meet 50–70% of the energy requirements of dairy cows [40,41]. Both dietary types and feed additives affect the rumen environment, including the number and activity of rumen microbiota, thereby changing the production of VFAs [42,43]. The VFAs produced by the rumen microbiota mainly include acetate, propionate, and butyrate [44]. Numerous studies have shown that butyrate promoted the development of the rumen, as it was converted into β-hydroxybutyrate and acetoacetate in the rumen epithelium and used as a direct energy donor for rumen epithelial cells and papilla development [45,46]. In the rumen of young ruminants, administration of any of the three main VFAs can promote the growth and functional maturation of rumen epithelium, with butyrate having the most remarkable effect on rumen papillae proliferation [47,48,49,50]. Previous studies have shown that butyrate was the most effective inducer of cellular functional changes in VFAs, which played an important role in regulating DNA histone modifications and gene networks, controlling cell differentiation and proliferation, inducing cell cycle arrest and apoptosis, and altering histone acetylation and methylation [51,52,53,54]. Available evidence has shown that iso-butyrate is essential for the development of the rumen in ruminants, which not only stimulates the growth of rumen organisms but promotes cellulose digestion [55]. As a branched-chain fatty acid (BCFA), iso-butyrate is typically present in the rumen as a metabolic product of protein bacterial catabolism [56]. In the rumen, fiber degradation in the diet produces acetate, and its linear increase in concentration may also be related to the improvement of digestion of structural carbohydrates, while fiber-degrading bacteria and starch-degrading bacteria increase the deposition of propionate. Different types of rumen fermentation have different A/P ratios. The higher the proportion of acetate, the more inclined the fermentation type is towards the acetate type, while the higher the proportion of propionate, the more inclined the fermentation type is towards the propionate type [57]. In the present study, supplementation of BA linearly increased the content of butyrate, iso-butyrate, and A/P ratio, indicating that BA might have the potential to promote rumen development in weaned Holstein dairy calves and to improve the digestion of structural carbohydrates.

The balance of the ecological environment of rumen is critical for ruminant nutrition, and the rumen’s ecological environment may be influenced by factors such as feeding methods, diet, and environment [57]. *Bifidobacterium*, an anaerobic probiotic that colonizes the intestines of humans and animals, participates in various physiological processes such as immunity, digestion, and absorption, playing an important role in maintaining the balance of the gut microbiota, inhibiting pathogen growth, and promoting gastrointestinal health [58]. Members of *Bifidobacterium* can evade rumen degradation through various dietary carbohydrates, a large amount of which typically helps promote intestinal and host health [59,60]. *Gastranaerophilales* (which belongs to the phylum *Cyanobacteria*), an early microbial biomarker reflecting the changes of flora in the gastrointestinal tract, is related to inflammatory response. As a fermentation bacterium, it may produce formate, lactate, ethanol, and CO_2_ under anaerobic conditions [61,62,63]. Wang et al. [64] found that *Gastranaerophilales* and leukotriene B_4_ (LTB_4_) were significantly increased in milk from clinical mastitis (CM) cows, and *Gastranaerophilales* was positively correlated with LTB_4_. Bach et al. [65] found that the relative abundance of *Gastranaerophilales* in the rumen was negatively correlated with milk yield. Perea et al. [66] found that *Gastranaerophilales* was associated with poor growth performance in lambs. In addition, previous research indicated that *Oscillospiraceae_UCG-002* was associated with abnormal health conditions [67,68]. Previous research also showed that when stocking density increased, *Oscillospiraceae_UCG-002* significantly enriched in the gut of hens, which might pose a threat to host health [69,70]. However, there are few studies which have reported the effect of BA on rumen microbiota of weaned Holstein dairy calves. In the present research, with the supplementation of BA, the relative abundance of *Bifidobacterium* significantly increased, but those of *Gastranaerophilales* and *Oscillospiraceae_UCG-002* significantly decreased. This might stimulate the rumen development of calves, improve rumen fermentation parameters, and promote their health, but the mechanism still needs further exploration.

The glyoxylate cycle and TCA cycle are sub-pathways of GLYCOLYSIS-TCA-GLYOX-BYPASS and GLYCOL-GLYOXDEG-PWY, and the latter is a variant of P105-PWY [71]. The up-regulation of the glyoxylate cycle is consistent with the fact that butyrate degradation leads to the production of two molecules of acetyl-CoA [72]. Indeed, the strong up-regulation also exists in enzymes (such as catalase) involved in antioxidant activity and cofactor synthesis [73]. Lacroux et al. [73] found that in the peroxisomes, butyrate underwent assimilation reaction, producing acetyl-CoA through the β-oxidation pathway and further producing tri/dicarboxylic acid in the glyoxylate cycle. Within the TCA cycle, butyrate has been shown to be the more effective oxidation substrate compared to ketone 3-hydroxybutyrate (3-OHB) [74]. Sanchez et al. [75] found that butyrate was decomposed into acetyl-CoA by the β-oxidation pathway, and then entered the TCA cycle and served as an energy substrate to produce citrate. Mathewson et al. [76] proved that most carbon from butyrate was bound into the TCA cycle, leading to the rapid metabolism of butyrate. At present, the specific functions of *Gastranaerophilales* are not clear, but studies have shown that it promotes host digestion and serves as a source of vitamins B and K [77]. Soo et al. [62] found that *Gastranaerophilales* only supported metabolism through fermentation because it seemed to lack genes for anaerobic and aerobic respiration. In the present research, *Gastranaerophilales* was positively correlated with the GLYCOLYSIS-TCA-GLYOX-BYPASS, GLYCOL-GLYOXDEG-PWY, and P105-PWY pathways, and negatively correlated with butyrate. Additionally, the GLYCOLYSIS-TCA-GLYOX-BYPASS, GLYCOL-GLYOXDEG-PWY, and P105-PWY pathways were negatively correlated with butyrate. With the supplementation of BA, the relative abundance of *Gastranaerophilales* decreased, the GLYCOLYSIS-TCA-GLYOX-BYPASS, GLYCOL-GLYOXDEG-PWY, and P105-PWY pathways were down-regulated, and the yield of butyrate increased. The relationship between the above three pathways and butyrate was consistent with previous research [71,72,74], while further research is needed to confirm the relationship between *Gastranaerophilales* and the pathways.

## 5. Conclusions

The present research indicated that supplementation with BA helps to promote the grow performance and improved rumen fermentation and the composition and function of rumen microbiota of weaned Holstein dairy calves. The addition of different doses of BA linearly increased the ADG and quadratically enhanced the ADFI and FBW, but linearly decreased the F/G ratio. Simultaneously, increasing doses of BA supplementation promoted rumen fermentation by linearly increasing the production of butyrate and iso-butyrate and the A/P ratio. In the rumen, the relative abundance of *Bifidobacterium* was increased, but those of the harmful bacteria, such as *unclassified_o__Gastranaerophilales*, and *Oscillospiraceae_UCG-002* were decreased. Finally, supplementation with different doses of BA down-regulated the GLYCOLYSIS-TCA-GLYOX-BYPASS, GLYCOL-GLYOXDEG-PWY, and P105-PWY pathways, which are associated with glycolysis. Under the conditions of this research, supplementation of 0.50% BA was optimal, which might be used as an effective method in the efficient breeding of weaned calves.

## Figures and Tables

**Figure 1 animals-14-02823-f001:**
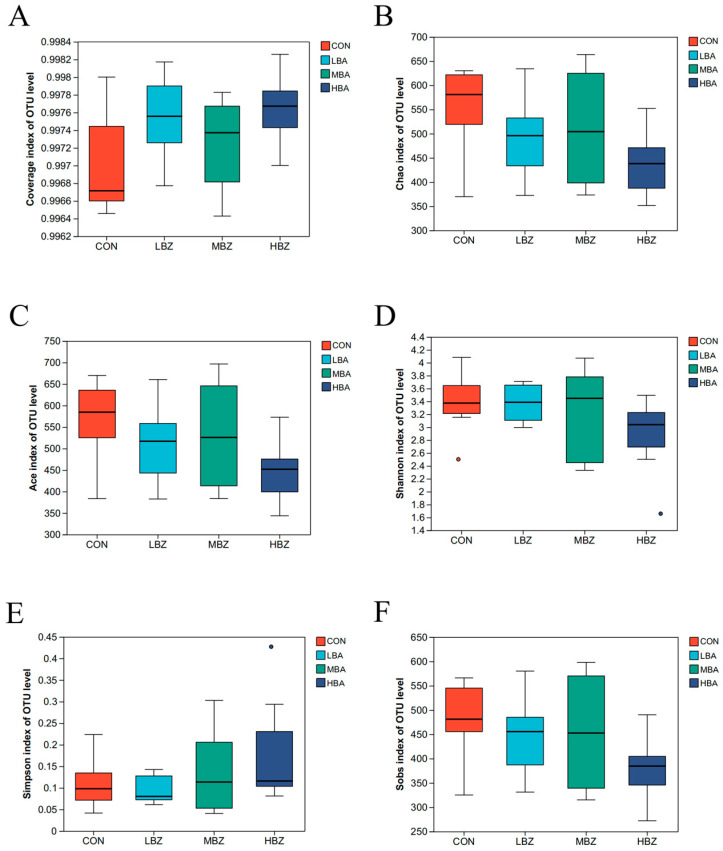
Alpha diversity analysis of rumen microbiota in weaned Holstein dairy calves. (**A**) Coverage index. (**B**) Chao index. (**C**) Ace index. (**D**) Shannon index. (**E**) Simpson index. (**F**) Sobs index. CON, the control group, which was fed with basal diet; LBA (low BA group), MBA (medium BA group), and HBA (high BA group) are the experimental groups, which were supplemented with 0.25%, 0.50%, and 0.75% BA in the basal diet (as dry matter basis), respectively.

**Figure 2 animals-14-02823-f002:**
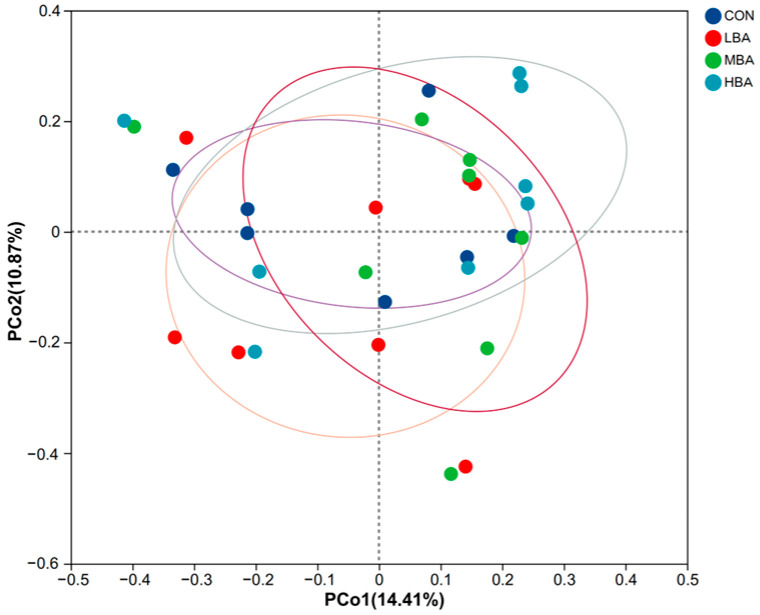
Principal coordinate analysis (PCoA) of rumen microbiota in weaned Holstein dairy calves based on Bray–Curtis dissimilarity. CON, the control group, which was fed with basal diet; LBA (low BA group), MBA (medium BA group), and HBA (high BA group) are the experimental groups, which were supplemented with 0.25%, 0.50%, and 0.75% BA in the basal diet (as dry matter basis), respectively.

**Figure 3 animals-14-02823-f003:**
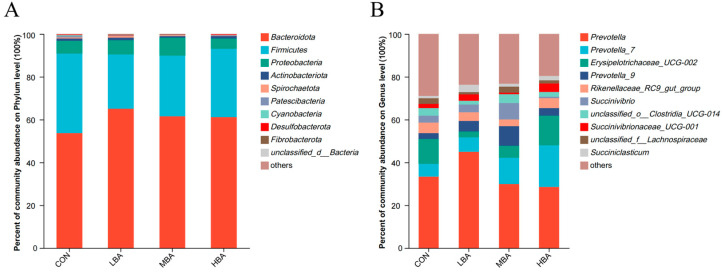
Relative abundances of rumen microbiota at phylum and genus levels in weaned Holstein dairy calves. (**A**) Phylum level. (**B**) Genus level. CON, the control group, which was fed with basal diet; LBA (low BA group), MBA (medium BA group), and HBA (high BA group) are the experimental groups, which were supplemented with 0.25%, 0.50%, and 0.75% BA on the basal diet (as dry matter basis), respectively.

**Figure 4 animals-14-02823-f004:**
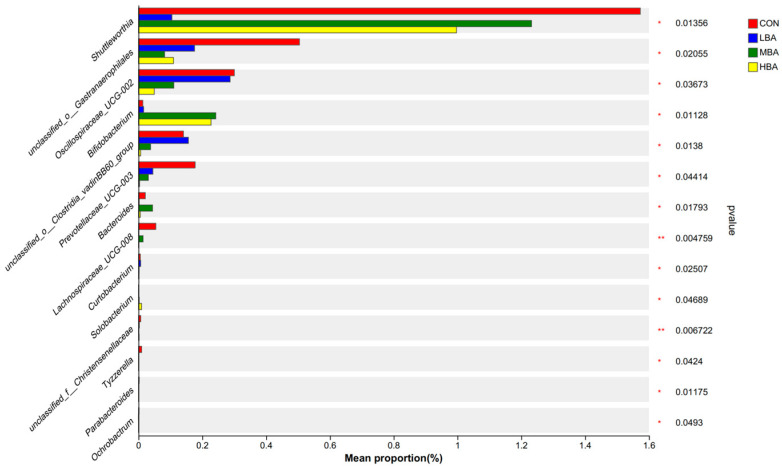
Comparison of rumen bacteria at the genus level in weaned Holstein dairy calves. CON, the control group, which was fed with basal diet; LBA (low BA group), MBA (medium BA group), and HBA (high BA group) are the experimental groups, which were supplemented with 0.25%, 0.50%, and 0.75% BA on the basal diet (as dry matter basis), respectively. * 0.01 < *p* < 0.05; ** 0.001 < *p* < 0.01.

**Figure 5 animals-14-02823-f005:**
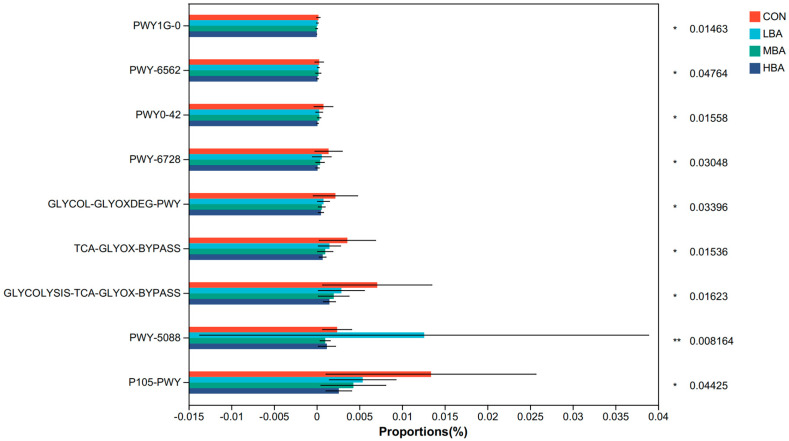
Functional changes in rumen microbiota of weaned Holstein dairy calves. CON, the control group, which was fed with basal diet; LBA (low BA group), MBA (medium BA group), and HBA (high BA group) are the experimental groups, which were supplemented with 0.25%, 0.50%, and 0.75% BA in the basal diet (as dry matter basis), respectively. * 0.01 < *p* < 0.05; ** 0.001 < *p* < 0.01.

**Figure 6 animals-14-02823-f006:**
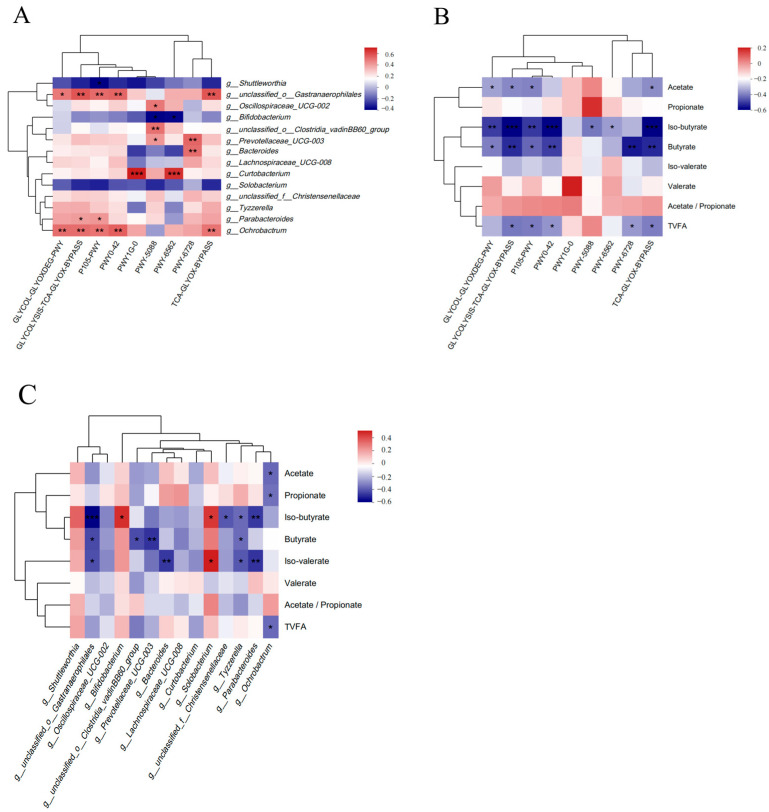
Correlation analysis. (**A**) Correlation analysis of differential pathways and differential rumen microbiota; (**B**) correlation analysis of differential pathways and VFAs; (**C**) correlation analysis of differential rumen microbiota and VFAs. The red represents positive correlation, blue represents negative correlation, and the depth of the color represents the degree of correlation. * 0.01 < *p* < 0.05; ** 0.001 < *p* < 0.01, *** *p* ≤ 0.001.

**Table 1 animals-14-02823-t001:** Diet composition and nutrient levels.

Items	Proportion (%)	
	Starter	Growth Feed
Diet composition (as-fed basis)		
Corn	32.00	39.73
Soybean Meal	17.00	9.73
Cottonseed Meal	2.73	3.27
DDGS	13.18	13.63
Bran	10.55	16.09
Puffed Soybean Flour	2.73	
Wheat Flour	4.54	
NaHCO_3_	0.45	0.73
^1^ Premix	7.73	7.73
Oat Grass	9.09	9.09
Total	100	100
Nutrient Levels (as dry matter basis)		
Dry Matter	89.03	88.64
Crude Protein	19.13	17.63
Ether Extract	2.88	2.48
Ash	9.47	6.40
Neutral Detergent Fiber	27.92	34.57
Acid Detergent Fiber	13.36	13.27
Ca	1.10	0.97
P	0.73	0.71

^1^ Supplied per kilogram of diet: Vitamin A 800 kIU, Vitamin D_3_ 240 kIU, Vitamin E 7 kIU, Vitamin C 200 mg, Vitamin K_3_ 15 mg, Zn 11,000 mg, Mn 7000 mg, Cu 1500 mg, I 40 mg, Se 25 mg, Co 15 mg.

**Table 2 animals-14-02823-t002:** Effect of different levels of BA on the growth performance in weaned Holstein dairy calves.

Items	Treatment ^1^				SEM ^2^	*p*-Value		
	CON	LBA	MBA	HBA		Treatment	Linear	Quadratic
IBW ^3^ (d 60, kg)	91.0	93.1	91.4	91.9	0.6	0.68	0.83	0.58
FBW ^4^ (d 102, kg)	137.1	146.6	144.6	140.9	1.4	0.08	0.56	0.02
ADG ^5^ (kg/d)	1.07 ^b^	1.22 ^a^	1.27 ^a^	1.24 ^a^	0.03	0.03	0.02	0.06
ADFI ^6^ (kg/d DM)	3.05 ^b^	3.30 ^a^	3.34 ^a^	3.18 ^ab^	0.03	0.01	0.27	0.02
F/G ratio ^7^	2.85	2.70	2.63	2.56	0.05	0.17	0.03	0.64

^1^ CON, the control group, which was fed with the basal diet; LBA (low BA group), MBA (medium BA group), and HBA (high BA group) are the experimental groups, which were supplemented with 0.25%, 0.50%, and 0.75% BA in the basal diet (as dry matter basis), respectively. ^2^ SEM, standard error of the mean; ^3^ IBW, initial body weight; ^4^ FBW, final body weight; ^5^ ADG, average daily gain; ^6^ ADFI, average daily feed intake; ^7^ F/G ratio, feed-to-gain ratio. ^a b^ Within a row, means with different superscripts differ, *p* < 0.05.

**Table 3 animals-14-02823-t003:** Effect of different levels of BA on rumen fermentation parameters in weaned Holstein dairy calves.

Items	Treatment ^1^				SEM ^2^	*p*-Value		
	CON	LBA	MBA	HBA		Treatment	Linear	Quadratic
VFAs ^3^ (mmol/L)								
Acetate	49.20	49.54	52.12	52.90	1.77	0.86	0.41	0.95
Propionate	32.26	28.28	28.82	30.67	1.85	0.94	0.97	0.54
Iso-butyrate	0.46 ^b^	0.71 ^a^	0.83 ^a^	0.89 ^a^	0.05	<0.01	<0.01	0.26
Butyrate	6.90 ^b^	7.89 ^ab^	8.56 ^ab^	10.52 ^a^	0.47	0.04	<0.01	0.58
Iso-valerate	0.87	1.09	1.22	1.19	0.12	0.76	0.34	0.62
Valerate	1.83	1.98	2.17	1.80	0.15	0.82	0.95	0.40
Acetate/Propionate	1.43	1.72	1.83	1.81	0.06	0.07	0.02	0.16
TVFAs ^4^ (mmol/L)	91.33	89.48	93.82	98.26	3.85	0.86	0.50	0.70

^1^ CON, the control group, which was fed with the basal diet; LBA (low BA group), MBA (medium BA group), and HBA (high BA group) are the experimental groups, which were supplemented with 0.25%, 0.50%, and 0.75% BA in the basal diet (as dry matter basis), respectively. ^2^ SEM, standard error of the mean; ^3^ VFAs, volatile fatty acids; ^4^ TVFAs, total volatile fatty acids. ^a b^ Within a row, means with different superscripts differ, *p* < 0.05.

## Data Availability

The 16S rRNA sequencing data for all the samples were deposited into the NCBI Sequence Read Archive (SRA) under accession number PRJNA1128666 (https://www.ncbi.nlm.nih.gov/sra/PRJNA1128666, accessed on 26 June 2024).

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
