# Peer review of "Effect of Dietary Benzoic Acid Supplementation on Growth Performance, Rumen Fermentation, and Rumen Microbiota in Weaned Holstein Dairy Calves"

_animals, 2024, doi:10.3390/ani14192823_

Round 1

Reviewer 1 Report

Comments and Suggestions for Authors

This study aimed to use organic compound, benzoic acid (BA), as an antimicrobial agent improving growth performance, rumen fermentation and microbiota in weaned dairy calves.

It was obvious that BA may enhance daily gain and boost Bacteroidetes community, resulting in a linear increase in butyrate acid production. The authors concluded that BA supplementation at 0.5% was optimal; nevertheless, F/G ratio in the 0.75%BA was considerably lower than 0%BA group (based on SEM shown), and butyrate acid production increased linearly as BA level increased. Please reconsider this conclusion.

The reference section contains no references to NRC (2012).

Author Response

Comments 1: This study aimed to use organic compound, benzoic acid (BA), as an antimicrobial agent improving growth performance, rumen fermentation and microbiota in weaned dairy calves. It was obvious that BA may enhance daily gain and boost Bacteroidetes community, resulting in a linear increase in butyrate acid production. The authors concluded that BA supplementation at 0.5% was optimal; nevertheless, F/G ratio in the 0.75% BA was considerably lower than 0% BA group (based on SEM shown), and butyrate acid production increased linearly as BA level increased. Please reconsider this conclusion.

Response 1: Thanks for your valuable suggestions. There was a significant difference in ADG on treatment, while there was no difference in F/G ratio, and the 0.5% BA group has the highest ADG. In addition, a dosage of 0.50% has lower costs and better economic benefits. Therefore, we believed that BA supplementation at 0.5% was optimal.

Comments 2: The reference section contains no references to NRC (2012).

Response 2: Thanks for reminding us. Here we used NRC (2021), which has been listed in the reference section (L125, L498).

Reviewer 2 Report

Comments and Suggestions for Authors

The manuscript Effect of Dietary Benzoic Acid Supplementation on Growth  Performance, Rumen Fermentation and Rumen Microbiota in Weaned Holstein Dairy Calve

aimed to investigate the effects of varying doses of BA supplementation on the growth performance, rumen fermentation, and rumen microbiota of weaned Holstein dairy calves to optimize the rearing practices.

The manuscript is well organized. A few minor suggestions for authors are included in the file attached.

Author Response

Comments 1: Add....to the basal diet (DM basis).

Response 1: Thank you for your comments. The manuscript has been revised, as suggested (L30-31).

Comments 2: It is contradictory. After weaning there are 35 days if the trial ended at 102 days.

Response 2: Thank you for your comment. We have explained the starting age of the experiment (L31).

Comments 3: Explain acronyms.

Response 3: Thanks for your valuable suggestions. We have adopted your suggestion (L36).

Comments 4: Try not to repeat the title words.

Response 4: Thanks for your valuable suggestions. We have made modifications to this section (L49).

Comments 5: Add for analysis.

Response 5: Thanks for reminding us. We have adopted your suggestion (L134).

Comments 6: No letters were reported to see the differences of the F/G ratio.

Response 6: As shown in Table 2 (L214), the P-value of the F/G ratio was 0.03 on linear. So, we described “the F/G ratio linearly decreased with the supplementation with BA (P = 0.03)”. In addition, we have also demonstrated the significance on treatment in the next sentence. (L211).

Comments 7: No statistical differences of A/P ratio were presented in the table.

Response 7: As shown in Table 3 (L221), the P-value of the A/P ratio was 0.02 on linear. In addition, we have also demonstrated the significance on treatment in the next sentence (L234-235).

Reviewer 3 Report

Comments and Suggestions for Authors

Dear Authors,

Please review the attached comments and address them carefully. 

Author Response

Major issues

Comments 1: Introduction. Needs firm justification. Hypothesis is required.

Response 1: Thank you for your good suggestion. We have revised the introduction section and added hypothesis in the revised manuscript (L51-94, L456-463).

Comments 2: Stat Analysis. Needs more information.

Response 2: The statistical analysis section listed the analysis methods, software, or models required for different data, while the analysis of microbial data was described in more detail in section 2.2.5 (L159-192).

Comments 3: Results. High-resolution images with clear labels are recommended.

Response 3: Thank you for your valuable suggestions. We have adjusted the layout inside Figure 6 and increased its resolution to make it clearer, as suggested (L315).

Comments 4: Weak Discussion and Conclusion. The discussion fails to connect the findings to existing literature, and some conclusions are not fully supported by the data. Additionally, the study's limitations are not adequately addressed.

Response 4: Thank you for your valuable comments. We think highly of your comments and suggestions. Now we have revised the discussion and conclusion sections, as suggested (L321-338, L422-434).

Comments 5: Inadequate References. There are missing citations for key points, thus reducing the credibility and relevance of the research.

Response 5: Thank you for your valuable comments. At present, the application of BA is limited, mostly focusing on monogastric. To our knowledge, there are few articles concerning the effects of BA on the growth performance of weaning calves. Therefore, in the first paragraph of the discussion section, we only listed one article about cattle (L333-335).

Other issues

Comments 1: Provide a simple summary for the general audience (L14).

Response 1: Thank you for your comments. We have revised this section, as suggested (L14-23).

Comments 2: Seems the calves are not weaned at the time of study started. So, you cannot claim that these are weaned at the time of study as you mentioned in the title (L31).

Response 2: Thank you for your comments. The dairy calves started weaning at 60 days of age, involving a 7-day transitional period during which milk feeding gradually decreased by 1 L per day until its cessation at 67 days of age. So, we claimed that these are weaned as we mentioned in the title.

Comments 3: Better not to use P values in the abstract (L33).

Response 3: Thank you for your suggestions. We have deleted the P values of abstract section, as suggested (L24-48).

Comments 4: Please check the following manuscript which has gone through the studies of weaning stresses of dairy calves. Try to grab some out of it for your introduction (L58).

(https://doi.org/10.3168/jds.2021-21009, https://doi.org/10.3390/ani12121474)

Response 4: Thank you for your comments. We have cited these two articles in the manuscript (L51-58, L456-459).

Comments 5: When you cite something that happened in the past, please use past tense rather than the present tense (L69)

Response 5: Thanks for reminding us. We have revised the grammatical error, as suggested (L70).

Comments 6: Please elaborate abbreviations when you first use them (ADG = Average Daily Gain) (L72). Check elsewhere.

Response 6: Thanks for reminding us. We have elaborated abbreviations that we ignored (L73, L201).

Comments 7: Do not start a sentence with an abbreviation (L79). Check elsewhere.

Response 7: Thanks for reminding us. The manuscript has been revised, as suggested (L80, L109, L168, L176, L272, L285, L322).

Comments 8: Livestock means it includes ruminants as well. Thus, use of such claims is invalid here. Thus, revise for more conciseness (L86).

Response 8: Thank you for your comments. The introduction section has been revised as you suggested. Now the original sentence has been removed (L87-90).

Comments 9: Heifers or bull calves (L99)?

Response 9: Thanks for reminding us. We used heifers in our experiment, which has now been explained in the manuscript (L102).

Comments 10: Please provide information about how these calves were raised during the first 60 days of their life. Colostrum feeding, housing, milk feeding volumes and other practices. Also, please mention how these calves were housed during the experiment. Single pens or group pens. Bedding arrangements (L107).

Response 10: Thanks for your valuable suggestion. We have introduced the feeding situation of calves for the first 60 days and explained the arrangement of calf pens during the experiment (L110-115).

Comments 11: What is growth feed? Please explain (L113).

Response 11: This solid feed is used to meet the growth and nutritional needs of weaned calves, and was fed to calves aged 68 to 180 days on the dairy farm where we conducted the experiment (L117-118). This kind of solid feed has been reported in previous publication (Abdelsattar et al., 2021).

Reference

Abdelsattar MM, Vargas-Bello-Pérez E, Zhuang Y, Fu Y, Zhang N. Impact of dietary supplementation of β-hydroxybutyric acid on performance, nutrient digestibility, organ development and serum stress indicators in early-weaned goat kids. Anim Nutr. 2021 9:16-22. doi: 10.1016/j.aninu.2021.11.003.

Comments 12: You already have mentioned ad libitum availability of feed and water. So, mentioning it again and again is not necessarily required (L114).

Response 12: Thanks for your valuable suggestion. We have removed the duplicate parts and only kept one (L122-123).

Comments 13: Any particular reason only to consider IBW and FBW only? it would have been better if weekly or biweekly BW was measured (L120).

Response 13: To avoid stress on calves, we only measured IBW and FBW to calculate ADG.

Comments 14: Why only one sampling at the end? How do you conclude that there is a difference compared to the day you started the experiment (L136)?

Response 14: The focus of our experiment was to compare the changes in growth performance, rumen fermentation, and rumen microbiota of calves fed with BA compared to the CON group. So, we only conducted one sampling at the end to compare the differences between groups.

Comments 15: Please explain the stat analysis in a more comprehensive manner mentioning the model used for the analysis (L185).

Response 15: We have explained the statistical analysis methods and software or models used for each section of the data, and the analysis of microbiota was further elaborated in section 2.2.5 (L159-192).

Comments 16: Please provide exact P values rather than saying < 0.05 or > 0.05. Check elsewhere (L196).

Response 16: Thanks for your commnets. The change has been made, as suggested (L206-212, L233-235, L273-275).

Comments 17: It seems treatment abbreviated in wrong manner. In figure 1 caption indicated CON, LBA, MBA, and HBA, but in the figure 1 itself says CON, LBZ, MBZ, and HBZ. Why? This is applicable to figure 2, 3 and 4 and so on as well.

Response 17: Thanks for reminding us. The manuscript has been revised, as suggested (Table 1 and 2, Figure 1, 2, 3, 4 and 5).

Comments 18: Check the spaces between words (L250).

Response 18: Thanks for reminding us. We have checked and there were no issues with the spaces between words (L258).

Comments 19: Why did you put an asterisk (L272)?

Response 19: We used asterisks to represent differences in figures, so we explained the differences represented by different asterisks here (L277, L280).

Comments 20: Figure 6 is not clear or visible to the naked eye.

Response 20: Thanks for reminding us. We have revised the Figure 6 to make it clearer, as suggested (L315).

Comments 21: Provide reference? Not the one from poultry or swine (L320).

Response 21: Although numerous studies indicated that supplementation with BA could improve the growth performance of livestock and poultry, few studies have reported the effect of BA on the growth performance of weaned Holstein dairy calves. We just found one document exploring the effects of BA on the growth performance of ruminants, which was listed in line 332-335.

Comments 22: This statement is partially correct. But the whole statement is not based on the results (L327).

Response 22: Thank you for your comments. The manuscript has been revised, as suggested (L336).

Comments 23: What is etc here? Mention all of them (L332).

Response 23: Thanks for reminding us. We have added “methane” as suggested (L341).

Comments 24: What is A/P (L353)?

Response 24: The A/P means Acetate/Propionate, as explained in line 233.

Comments 25: Mention all of them (L381).

Response 25: This sentence has already mentioned all the microbiota listed in the discussion section (L391).

Comments 26: I do not get the meaning of this statement (L419).

Response 26: Thanks for reminding us. We have revised the corresponding statement, as suggested (L432-434).
